# Efficacy of ChAdOx1 nCoV-19 (AZD1222) vaccine against SARS-CoV-2 lineages circulating in Brazil

Sue Ann Costa Clemens[1,2,33], Pedro M. Folegatti [3,33], Katherine R. W. Emary[1,33], Lily Yin Weckx[4,33], Jeremy Ratcliff [5,33], Sagida Bibi[1,33], Ana Verena De Almeida Mendes[6,7], Eveline Pipolo Milan[8], Ana Pittella[9,10,11], Alexandre V. Schwarzbold[12], Eduardo Sprinz[13,14], Parvinder K. Aley[1], David Bonsall[15], Christophe Fraser [15], Michelle Fuskova[3], Sarah C. Gilbert [3], Daniel Jenkin[3], Sarah Kelly[1], Simon Kerridge[1], Teresa Lambe[3], Natalie G. Marchevsky[1], Yama F. Mujadidi[1], Emma Plested[1], Maheshi N. Ramasamy[1,16], Peter Simmonds[5], Tanya Golubchik [17,34], Merryn Voysey [1,34✉], Andrew J. Pollard [1,34✉], the AMPHEUS Project* & Oxford COVID Vaccine Trial Team*

Several COVID-19 vaccines have shown good efficacy in clinical trials, but there remains uncertainty about the efficacy of vaccines against different variants. Here, we investigate the efficacy of ChAdOx1 nCoV-19 (AZD1222) against symptomatic COVID-19 in a post-hoc exploratory analysis of a Phase 3 randomised trial in Brazil (trial registration ISRCTN89951424). Nose and throat swabs were tested by PCR in symptomatic participants. Sequencing and genotyping of swabs were performed to determine the lineages of SARS-CoV-2 circulating during the study. Protection against any symptomatic COVID-19 caused by the Zeta (P.2) variant was assessed in 153 cases with vaccine efficacy (VE) of 69% (95% CI 55, 78). 49 cases of B.1.1.28 occurred and VE was 73% (46, 86). The Gamma (P.1) variant arose later in the trial and fewer cases ($N = 18$) were available for analysis. VE was 64% (−2, 87). ChAdOx1 nCoV-19 provided 95% protection (95% CI 61%, 99%) against hospitalisation due to COVID-19. In summary, we report that ChAdOx1 nCoV-19 protects against emerging variants in Brazil despite the presence of the spike protein mutation E484K.

[1] Oxford Vaccine Group, Department of Paediatrics, University of Oxford, Oxford, UK. [2] Institute of Global Health, University of Siena, Siena, Italy. [3] The Jenner Institute, Nuffield Department of Medicine, University of Oxford, Oxford, UK. [4] Department of Pediatrics, Universidade Federal de São Paulo, São Paulo, Brazil. [5] Nuffield Department of Medicine, Peter Medawar Building for Pathogen Research, University of Oxford, Oxford, UK. [6] Escola Bahiana de Medicina e Saúde Pública, Brazil and ID'OR, Salvador, Brazil. [7] Hospital São Rafael, Salvador, Brazil. [8] Universidade Federal do Rio Grande do Norte - UFRN, Natal, Brazil. [9] Hospital Quinta D'Or, Rio de Janeiro, Brazil. [10] Instituto D'Or de Pesquisa e Ensino (IDOR), Rio de Janeiro, Brazil. [11] Universidade Unigranrio, Rio de Janeiro, Brazil. [12] Clinical Research Unit, Department of Clinical Medicine, Universidade Federal de Santa Maria, Santa Maria, Brazil. [13] Infectious Diseases Service, Hospital de Clinicas de Porto Alegre, Porto Alegre, Brazil. [14] Universidade Federal do Rio Grande do Sul, Porto Alegre, Brazil. [15] Big Data Institute, Nuffield Department of Medicine, University of Oxford, Oxford, UK. [16] Oxford University Hospitals NHS Foundation Trust, Oxford, UK. [17] Oxford Viral Sequencing Group, Wellcome Centre for Human Genetics, University of Oxford, Oxford, UK. [33] These authors contributed equally: Sue Ann Costa Clemens, Pedro M. Folegatti, Katherine R. W. Emary, Lily Yin Weckx, Jeremy Ratcliff, Sagida Bibi. [34] These authors jointly supervised: Tanya Golubchik, Merryn Voysey, Andrew J Pollard. *A list of authors and their affiliations appears at the end of the paper. ✉email: merryn.voysey@paediatrics.ox.ac.uk; Andrew.pollard@paediatrics.ox.ac.uk

The SARS-CoV-2 pandemic continues to cause global impact. Spike protein-based SARS-CoV-2 vaccines have shown effectiveness in several countries[1,2] enabling the relaxation of non-pharmaceutical interventions in some settings. The emergence of SARS-CoV-2 variants prompts questions about the ongoing protection elicited by existing vaccines. The risk of vaccine escape, whereby vaccine generated immunity is insufficient to provide protection against disease, is a concern. High virus transmission in combination with the presence of convalescent or vaccine-mediated immunity may drive selection of escape mutants. These theoretical concerns are broadly supported by in vitro data showing reduction in neutralising antibody titres, but efficacy or clinical effectiveness of existing spike-based vaccines against the Alpha (B.1.1.7) variant of concern (VOC) does not seem to be compromised[3,4]. However, such vaccines have a reduced efficacy against the Beta (B.1.351) VOC that possesses additional spike mutations[5–7] and may translate into reduced vaccine effectiveness[8]. Nevertheless, several lines of evidence indicate that efficacy against severe disease may be preserved against current identified VOCs.

Brazil has experienced more than 16 million confirmed cases and over 450,000 deaths to date[9], with the Amazon region being particularly severely affected[10]. Lineages B.1.1.33 and B.1.1.28 were dominant throughout Brazil during 2020[11]. Towards the end of 2020, two sublineages of B.1.1.28, designated Zeta (P.2) and Gamma (P.1), emerged and spread rapidly through the population[11,12]. Prior infection with earlier lineages may not confer adequate or sustained protection in the face of emerging variants[13]. For example, areas in Brazil with suspected high seroprevalence rates have seen subsequent exponential growth of infections[10]. This contrasts with the protection from reinfection for a median of 7 months duration seen in a large healthcare worker (HCW) study in the UK during a period when B.1 lineages were circulating and the Alpha (B.1.1.7) variant arose[14]. Symptomatic reinfections in immunocompetent adults with Gamma (P.1) and Zeta (P.2) sublineages have been described (following B.1 and B.1.133 infections respectively)[15,16].

Both the Gamma (P.1) and Zeta (P.2) sublineages harbour the E484K mutation in the receptor binding domain (RBD) of the spike protein. E484K has been associated with in vitro immune escape from therapeutic monoclonal antibodies[17–19], prompting the withdrawal of the emergency use authorisation for bamlanivumab in the US[20]. The E484K mutation is observed to have arisen independently in other variants such as Beta (B.1.351)[21] and features as an additional mutation in recent samples of established VOCs such as Alpha (B.1.1.7)[22]. Whilst P.2 harbours no other lineage-specific spike mutations, Gamma (P.1) has additional RBD mutations, most notably K417T and N501Y. The coincident emergence of N501Y, K417T/N and E484K mutations in Gamma (P.1) and Beta (B.1.351)[21,23] is suggestive of convergent evolution[12].

The shared triplet of RBD mutations might suggest that the pattern of in vitro responses[24,25] and reduced efficacy of ChAdOx1 nCov-19[5] for Beta (B.1.351) may be echoed for Gamma (P.1). However, early in vitro data showed two monoclonal antibodies retained activity against Gamma (P.1) while showing no neutralization against Beta (B.1.351)[19,24]. Convalescent sera from individuals infected early in the pandemic and from mRNA and viral-vectored vaccine recipients showed a reduction in Gamma (P.1) neutralization activity for both pseudovirus and live coronavirus[19] but not to the extent seen for Beta (B.1.351)[26]. Early data for an inactivated vaccine in Brazil when Gamma (P.1) was dominant also suggest there may be a reduction in effectiveness[27].

In this paper, we report the findings from a multisite Brazilian COVID-19 vaccine efficacy study assessing the efficacy of the ChAdOx1 nCoV-19 vaccine in preventing symptomatic COVID-19 disease caused by the individual circulating SARS-CoV-2 lineages.

## Results

There were 10416 participants enroled and randomised into the study between June 23, 2020 and December 1, 2020. 9433 participants received two doses and met the criteria for inclusion in this analysis. Reasons for exclusion are shown in Fig. 1.

677 clinical samples were shipped and processed. Of these, 307 (45%) came from cases of primary symptomatic COVID-19 meeting the definition for inclusion in the efficacy analysis, and 236 (77%) of these primary cases had sufficient intact specimen for lineage assignment through sequencing or genotyping. Some participants had more than one positive swab for the same event.

Demographic and baseline characteristics of the primary efficacy cohort were well balanced (Table 1). A total of 82% of the participants were aged 18–55 years and 70% identified as white. A total of 65% worked in a health or social care setting.

The most prevalent lineage identified was Zeta (P.2) in 153 cases, followed by the ancestral B.1.1.28 lineage in 49 cases. Unblinding of study participants began at a similar time as the appearance of the Gamma (P.1) variant and only 18 cases were able to be included in the analysis (Fig. 2a, b). There were 46 cases of symptomatic NAAT + COVID-19 that occurred after a single dose or before the receipt of the second dose, including 22 cases of Zeta (P.2). These are summarised by lineage in Supplementary Table 4.

Vaccine efficacy after two doses was 73%, (95% CI, 46, 86) for B.1.1.28, and for Zeta (P.2) was 69% (95% CI, 55, 78). Fewer cases were available for analysis of efficacy for B.1.1.33 (VE 88.2%, 95% CI 5, 99), and Gamma (P.1) (64%, 95% CI, −2, 87) which had wide confidence intervals. Efficacy was not computed for cases of N.9 ($N = 4$), N.6 ($N = 1$) or Alpha (B.1.1.7) ($N = 1$) as there were fewer than 5 instances of each. Swabs that were not available for sequencing as the participant had accessed PCR testing at a non-study lab were imputed using a multiple imputation (MI) model. The MI analysis gave estimates of 65.8% 95% CI (5, 88) for Gamma (P.1) and 65.2% 95% CI (53. 74) for the Zeta (P.2) (Table 2).

Primary outcome hospitalisation cases, occurring more than two weeks after a second dose, were present in 1 and 18 participants in the ChAdOx1 nCoV-19 and control groups respectively, VE 95% (95% CI 61, 99). The one hospitalised participant in the ChAdOx1 nCoV-19 group had a WHO score of 5, but no swab was available for processing to determine lineage. There were no severe cases nor deaths in the vaccinated arm. Among the participants meeting the criteria for primary efficacy analysis, there was one death due to COVID-19 in the control arm, and 6 further cases were classified as severe COVID-19 (WHO score ≥ 6), also in the control arm, giving 100% efficacy (95% CI 34, NE) against severe COVID-19 with two doses of vaccine (Table 3, Supplementary Table 1). A second death occurred in the control arm more than 21 days after the first dose of vaccine but before the second dose was received.

Viral load varied by SARS-CoV-2 lineage ($p = 0.0002$) with the Gamma (P.1) lineage having the highest median viral load (Fig. 3, Supplementary Table 2).

## Discussion

In this post-hoc exploratory analysis, ChAdOx1 nCoV-19 provided protection against severe disease and death in Brazil, the key endpoints to protect lives and safeguard medical infrastructure from being overwhelmed. This analysis also shows vaccine efficacy against the dominant lineages causing

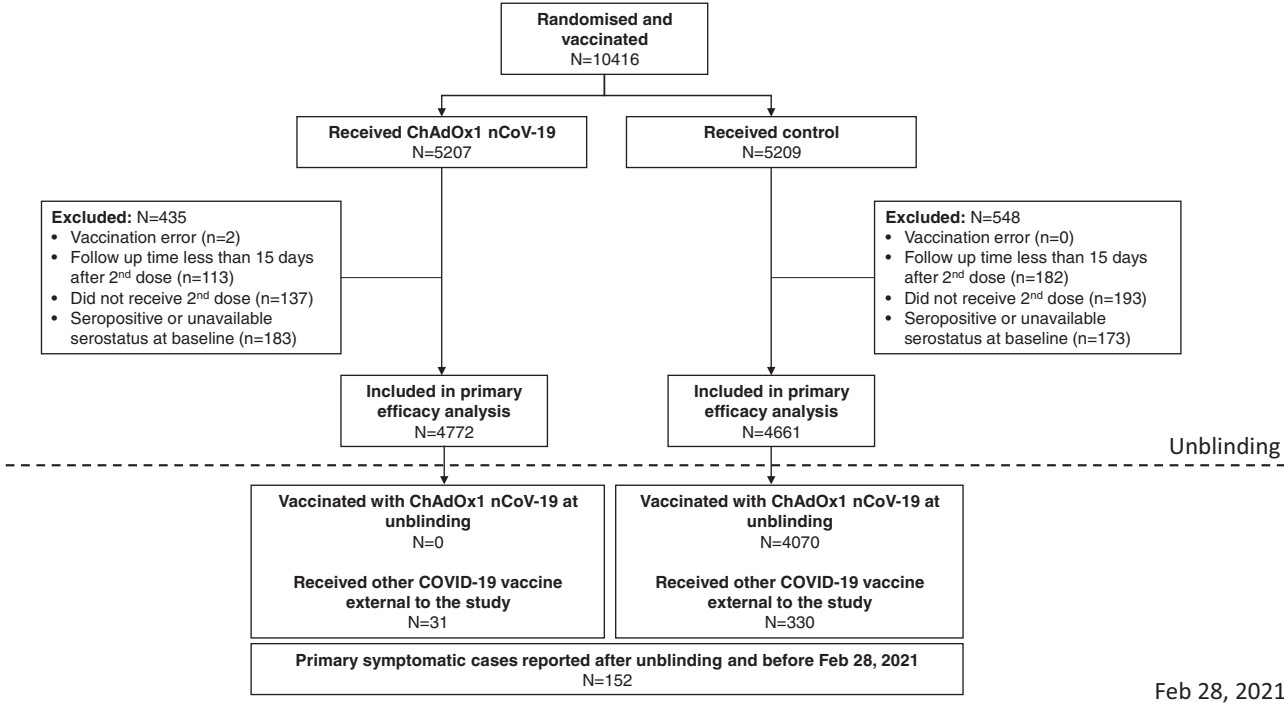

**Fig. 1 CONSORT Flow diagram.** Flow chart showing; the number of participants randomised and vaccinated with ChAdOx1 nCoV-19 or control vaccines; the number of participants included in the primary efficacy analysis and reasons for exclusion; the number of participants receiving vaccines after unblinding; and cases occurring after unblinding. Feb 28, 2021 was the data cut-off date for this analysis and events occurring after this date are not included in the data set for analysis.

symptomatic COVID-19 infection in our participants: Zeta (P.2) and B.1.1.28. There were relatively few cases of the B.1.1.33 and Gamma (P.1) lineages observed in the timeframe prior to unblinding, and assessment of efficacy for these variants was underpowered. The distribution of P.1 cases observed suggest that protection against symptomatic disease for this variant may be maintained but slightly reduced in comparison with Zeta (P.2) or the parent lineage B.1.1.28. However, the limited number of cases available for analysis makes it difficult to draw firm conclusions.

All first-generation spike-based COVID-19 vaccines that are currently in clinical use were generated from the ancestral Wu-1 *spike* gene sequence, raising the potential for loss of vaccine efficacy as SARS-CoV-2 accumulates mutations during viral evolution. Our observations of vaccine efficacy of ChAdOx1 nCoV-19 against symptomatic disease in this report are consistent with our primary combined analysis of efficacy from studies in Brazil, the UK, and South Africa, in which VE was 66·7% (57·4 to 74·0). The single-dose adenovirus vectored vaccine (Ad26.COV2.S) phase 3 data showed efficacy against moderate to severe-critical COVID-19 disease of 68.1% (95% CI, 48.8 to 80.7) where Zeta (P.2) formed the majority of the sequences obtained[28]. A recent pre-print of a test negative Canadian case control study showed vaccine effectiveness of 48% (28 to 63) >14 days post 1 dose of ChAdOx nCoV-19 against combined Beta (B.1.351)/ Gamma (P.1). Only single dose data were available given the timing of authorisation of ChAdOx1 nCoV-19 in Canada. This study had insufficient specimens to distinguish between these lineages and were thus combined, which emphasises the difficulty of achieving an adequate number of sequences for statistical comparison[29].

Our data are also in keeping with the high levels of protection against severe disease caused by other variants such as Beta (B.1.351) by BNT162b2 and NVX-CoV2373[30,31]. However, the positive findings from this study for Zeta (P.2) are in contrast to

the lack of observed efficacy seen for ChAdOx1 nCoV-19 against mild-moderate disease caused by Beta (B.1.351)[5]. There is a wide clinical spectrum of SARS-CoV-2 infection, from asymptomatic to severe COVID-19 disease requiring multi-organ support. The immune responses required to protect from asymptomatic disease may differ in nature or magnitude from those required to

**Table 1 Demographics and baseline characteristics of primary efficacy cohort.**

| Demographics | ChAdOx1 nCoV-19 All participants ($n = 4772$) | Control All participants ($n = 4661$) |
|---|---|---|
| Age | | |
| 18–55 years | 3854 (81%) | 3796 (81%) |
| 56–69 years | 765 (16%) | 735 (16%) |
| ≥70 years | 153 (3%) | 130 (3%) |
| Sex (female) *n%* | 2657 (56%) | 2508 (54%) |
| BMI (median, IQR) kg/m$^2$ | 26 [23, 29] | 26 [23, 29] |
| Ethnicity | | |
| White | 3299 (69%) | 3254 (70%) |
| Black | 410 (9%) | 409 (9%) |
| Asian | 124 (3%) | 98 (2%) |
| Mixed | 918 (19%) | 874 (19%) |
| Other | 11 (<1%) | 12 (<1%) |
| Missing | 10 (<1%) | 14 (<1%) |
| Health and social care setting workers *n%* | 3097 (65%) | 2974 (64%) |
| Co-morbidities | | |
| Cardiovascular disease | 798 (17%) | 782 (17%) |
| Respiratory disease | 491 (10%) | 448 (10%) |
| Diabetes | 231 (5%) | 185 (4%) |

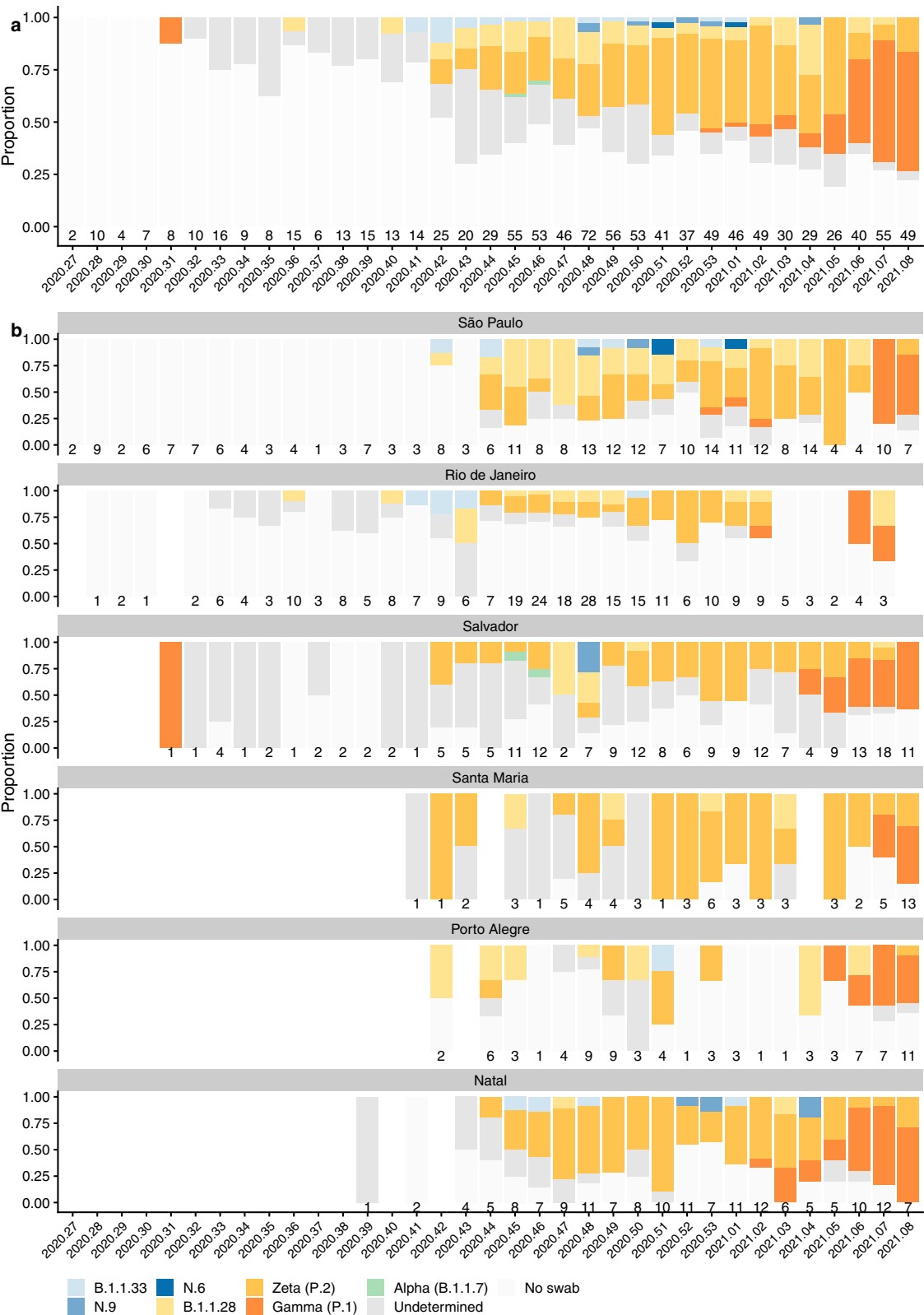

protect against severe disease which may in turn have implications for the ability of SARS-CoV-2 vaccines to reduce transmission. Animal data from the ChAdOx1 nCoV-19 vaccinated hamster model showed a reduction in virus neutralising antibody titre with Beta (B.1.351) compared with Alpha (B.1.1.7). However, when challenged with either of these lineages, the vaccinated animals did not have infectious virus or gross pathology in their

lungs yet virus detectable in the upper respiratory tract of both vaccinated and control animals[32].

Ongoing antigenic drift of the SARS-CoV-2 virus due to error-prone RNA replication is inevitable and it is possible that vaccines will drive the selection of variants towards escape from neutralising antibodies and to increased transmissibility. Many of the RBD mutations that have arisen appear to be associated with

**Fig. 2 Distribution of SARS-CoV-2 lineages from nose/throat swabs over time. a** Stacked bar chart of cases of NAAT + SARS-CoV-2 each week during the study, with lineage assigned by sequencing or genotyping where available. **b** Stacked bar chart of cases of NAAT + SARS-CoV-2 each week during the study, by study site, with lineage assigned by sequencing or genotyping where available. The 6 study sites are: Sao Paulo, Rio de Janeiro, Salvador, Santa Maria, and Porto Alegre. (see map of sites in Supplementary Fig. 3). X-axis labels show calendar year and week number. Numbers above the x-axis show the number of cases of NAAT + SARS-CoV-2 that occurred in the study during that week. Swabs were available for sequencing and genotyping only if participants were tested at a study site laboratory and the study sample was stored. An early sample from August 2020 was assigned to Gamma (P.1) to the presence of the K417T mutation. Phylogeographic analyses suggest emergence of the dominant P.1 lineage in November 2020, with a most recent common ancestor of all P.1-like (K417T) viruses estimated at August 2020[48]. As low viral load of this sample in our dataset precluded sequencing, we were unable to further refine its phylogenetic lineage. Therefore it is plausible that this sample was a precursor to likely 'true' Gamma (P.1) or a spontaneous K417T mutation. In keeping with national surveillance data, multiple instances of Gamma (P.1) samples were observed in our data from January 2021.

---

**Table 2 Efficacy of 2 doses of ChAdOx1 nCoV-19 against primary symptomatic COVID-19, by SARS-CoV-2 lineage.**

| Lineage | ChAdOx1 nCoV-19 n (%) N = 4772 | Control n (%) N = 4661 | Vaccine efficacy (95% CI) | Vaccine efficacy using multiple imputation for unavailable swabs (95% CI) |
|---|---|---|---|---|
| B.1.1.33 | 1 (0.0%) | 8 (0.2%) | 88.2 (5.4, 98.5) | |
| B.1.1.28 | 11 (0.2%) | 38 (0.8%) | 72.6 (46.4, 86.0) | |
| P.2 (Zeta) | 38 (0.8%) | 115 (2.5%) | 68.7 (54.9, 78.3) | 65.2 (52.5, 74.4) |
| P.1 (Gamma) | 5 (0.1%) | 13 (0.3%) | 63.6 (−2.1, 87.0) | 65.8 (4.9, 87.7) |
| Undetermined | 22 (0.5%) | 48 (1.0%) | 56.6 (28.2, 73.8) | |

*Undetermined lineage are those where a lineage could not be assigned due to low viral load or degraded RNA.

---

**Table 3 Hospitalisations (WHO score >= 4) by SARS-CoV-2 lineage and WHO clinical progression score.**

| WHO clinical progression score | | ChAdOx1 nCoV-19 | | | | Control | | | |
|---|---|---|---|---|---|---|---|---|---|
| | | 4 | 5 | 6 | 10 | 4 | 5 | 6 | 10 |
| **Secondary cases** (>21 days after dose 1, <15 days after dose 2) | Undetermined/no swab | 0 | 0 | 0 | 0 | 1 | 2 | 0 | 1 |
| | B.1.1.33 | 0 | 0 | 0 | 0 | 1 | 0 | 0 | 0 |
| | P.1 (Gamma) | 0 | 0 | 0 | 0 | 1 | 0 | 0 | 0 |
| | **Total** | **0** | | | | **6** | | | |
| **Primary cases** >=15 days after dose 2 | Undetermined/no swab | 0 | 1 | 0 | 0 | 3 | 3 | 4 | 1 |
| | B.1.1.28 | 0 | 0 | 0 | 0 | 0 | 3 | 1 | 0 |
| | P.1 (Gamma) | 0 | 0 | 0 | 0 | 0 | 1 | 0 | 0 |
| | P.2 (Zeta) | 0 | 0 | 0 | 0 | 1 | 0 | 1 | 0 |
| | **Total** | **1** | | | | **18** | | | |
| | VE 95% (95% CI 61%, 99%) | | | | | | | | |

*Undetermined lineage are those where a lineage could not be assigned due to low viral load or degraded RNA.
4 = Hospitalised; no oxygen therapy, 5 = Hospitalised; oxygen by mask or nasal prongs, 6 = Hospitalised; oxygen by NIV or high flow, 10 = dead.

---

immune evasion, transmissibility or both. The only RBD lineage defining mutation for Zeta (P.2) lineage is the E484K mutation, whilst Gamma (P.1) and Beta (B.1.351) harbour multiple RBD mutations. E484K (and a similar mutation E484Q) are being rapidly accumulated by lineages across distinct epidemiological and geographic settings and the addition of E484K/Q mutations to existing VOCs (such as Alpha (B.1.1.7)) is associated with evasion of neutralising antibodies[17,33]. The observation that vaccine efficacy in our trial was preserved for P.2 may indicate that E484K, when occurring as an isolated RBD mutation, may be responsible for minimal reduction in protection. However, it is not known what the relative contribution of E484K/Q mutations may have on vaccine efficacy when occurring as part of a constellation of RBD mutations. A cautious approach to variants containing E484K and other RBD mutations is warranted whilst our understanding of their individual impact improves.

The viral load was highest in the Gamma (P.1) cases consistent with other analyses[12,34]. Higher viral loads may result in more shedding of virus, contributing to the greater transmissibility seen with this variant[12]. It has been suggested that the time between onset of illness and NAAT testing might vary during the progression of the pandemic, confounding attempts to compare viral loads for different variants[35]. In our study there was a consistent median 4-day difference between illness onset and the collection date of the swab across all identified lineages thus comparisons of viral load are not confounded by this potential source of bias. Of note, samples with undetermined lineages had a larger median 8-day interval (IQR 6, 12) between illness onset and NAAT swab which may have resulted in the sample being taken at a time of reduced viral load making it more difficult to assign a lineage to the event.

The limitations of these data are that the sample size was determined by the number of samples from which a sequence sufficient to define lineage could be generated and was not sufficient to enable comparisons of efficacy between lineages. The evolution of the virus over time and between geographically distant trial sites resulted in a dataset with limited numbers in some lineage groups for efficacy analysis. Our study sites were

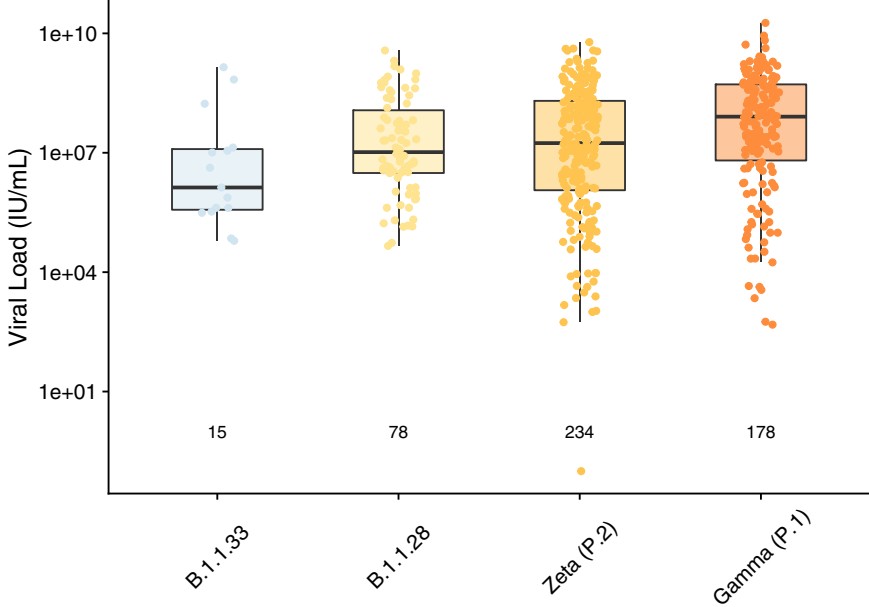

**Fig. 3 Viral Load in nose and throat swabs by SARS-CoV-2 lineage.** Box plot of viral load (IU/mL) from NAAT + SARS-CoV-2 cases in Brazil, in vaccinated and control participants combined. Lineages were assigned by sequencing and genotyping. Number of cases included in the analysis are shown below each box. Boxes show the median and 25th to 75th percentile range (bounds of the boxes) and whiskers to the last data point within 1.5 x interquartile range from the 25th or 75th percentile. Kruskal-Wallis test across all four groups: two-sided $p = 0.0002$. Different colours represent different lineages as labelled on the x-axis.

situated in the South and East of Brazil which may explain the relatively small proportion of Gamma (P.1) cases by the time of data cut-off. Phylogenetic evidence suggests this lineage arose in North West Brazil and a corresponding delay in observations from other parts of the country would be expected in line with epidemic spread[12]. In addition, the trial participants were unblinded as to allocation arm to allow participants to be vaccinated once efficacy was established, as requested by the ethics committee, thereby necessarily truncating the participants' ongoing inclusion for efficacy analysis. The unblinding of the study occurred at a time when Gamma (P.1) infections were growing rapidly in our study site areas. There were 18 cases of Gamma (P.1) included in the efficacy analysis and 160 that occurred after unblinding which could not be included in the efficacy analyses (supplementary table S2). However, every effort was made to assign a lineage for relevant samples obtained prior to unblinding by using a novel allele specific PCR method and missing data were imputed in a sensitivity analysis which yielded similar efficacy estimates to the complete case analysis. Our trial participants were also predominantly younger (<56 years) with relatively few co-morbidities, however despite this there was still evidence of protection against severe disease and death.

National roll-out of 2 COVID-19 vaccines, the Sinovac Biotech Ltd and Oxford/AstraZeneca vaccines, began in Brazil in January 2021, prior to study unblinding, and further vaccines have been subsequently approved for use. More than 20.1% of the population (total population ~212 million) had received at least one dose of a COVID-19 vaccine by 25th May 2021. Vaccine effectiveness studies are underway to evaluate real world impact on the pandemic in Brazil as vaccine trial efficacy of first-generation vaccines in most settings will no longer be attainable due to population vaccine roll out.

For next generation vaccines, studies to ascertain efficacy are likely to be based upon immunogenicity data showing equivalence to an as yet undefined immune correlate for protection which will be established from phase 3 trials. However, the variability of vaccine efficacy may be underpinned by genetic

mismatch between the vaccine lineage and currently circulating virus[36]. Defining the correct immune correlate is challenging in the face of continued antigenic drift, and selection pressure from previous infection and vaccine induced immunity. Work is ongoing to establish the role of variant vaccines, heterologous schedules and booster regimes.

The likelihood that vaccine effectiveness may vary against emerging SARS-CoV-2 variants emphasises the need for the infrastructure for ongoing viral genomic surveillance. This is particularly important in countries where both viral transmission is high and vaccination coverage is limited, and may need support from international agencies.

## Methods

**Overview.** An ongoing randomised controlled phase 3 multi-site trial of the efficacy of the ChAdOx1 nCoV-19 vaccine was conducted in Brazil that began on June 23, 2020. Efficacy, safety, and immunogenicity data, including the primary and secondary outcomes of the study, as well as the full study protocol have been previously published as part of a prespecified analysis of pooled data from UK, Brazil, and South Africa[37,38].

**Study design and participants.** This multi-centre study assessing the safety and efficacy of ChAdOx1 nCoV-19 vaccine was performed at six sites across Brazil (São Paulo, Rio de Janeiro, Salvador, Natal, Santa Maria, Porto Alegre) (Supplementary Fig. 3). Individuals aged 18 and over at high risk of exposure to SARS-CoV-2, with healthcare workers prioritised for enrolment.

Participants were screened for inclusion and exclusion criteria, underwent medical history review, clinical observations, history-directed clinical examination and provided informed consent.

Participants were randomised 1:1 using REDCap 10.6.13 to receive either ChAdOx1 nCoV-19 ($3.5–6.5 \times 10^{10}$ viral particles) or MenACWY conjugate vaccine as a control, administered as an intramuscular injection. Participants randomised to the control group received saline as their second dose. In response to emerging data from our phase 1 ChAdOx1 nCoV-19 study showing a rise in neutralising antibody with a second dose[39], all participants were offered a second dose with a dose interval of between 4 and 12 weeks (median 35 days, IQR 32, 47). Participants, clinical investigators and laboratory staff were blinded to vaccine allocation. Following emergency use authorisation of ChAdOx1 nCoV-19 and an inactivated SARS-CoV-2 viral vaccine in Brazil on 17th January 2021, all trial participants were unblinded to vaccine allocation but remained in the trial and continued with follow up. Participants in the control group were offered 2 doses of

ChAdOx1 nCoV-19 within the trial with a dose interval in line with the national programme or could choose to accept the inactivated viral vaccine as part of the Brazilian national immunisation programme.

Participants were asked to contact their study site if they developed any one of: fever of ≥37.8 °C, cough, shortness of breath or anosmia/ageusia. They were reminded weekly to do so throughout the trial. Symptomatic participants were invited for nasopharyngeal and oropharyngeal swabbing and a SARS-CoV-2 nucleic acid amplification test (NAAT) at their local clinical site. Samples were processed using commercial NAAT assays at local diagnostic laboratories listed in Supplementary Table S5. Swabs were shipped to Oxford for sequencing and genotyping as described in the supplementary methods section.

*Outcomes.* The primary objective of the trial was to evaluate efficacy of the ChAdOx1 nCoV-19 vaccine against NAAT-confirmed COVID-19. The primary outcome was virologically-confirmed, symptomatic COVID-19, defined as a NAAT-positive swab combined with at least one of: fever ≥37.8 °C, cough, shortness of breath, anosmia or ageusia. All NAAT positive cases occurring before participant unblinding and vaccination were reviewed by a blinded independent endpoint adjudication committee who assigned severity scores using the WHO clinical progression score[40]. Only cases adjudicated by the committee as primary outcome cases were included in the analysis. Participants continued to be followed up for SARS-CoV-2 infection after unblinding and subsequent vaccination and these cases were adjudicated by an internal adjudication committee and are not included in efficacy analyses.

Analysis by lineage is a post-hoc exploratory analysis.

*Statistical methods.* Participants were included in primary efficacy analyses if they were seronegative to the nucleocapsid protein at baseline, received two doses of vaccine, had follow up for at least 15 days after the second dose, and no prior evidence of infection. Cases were included in the efficacy analysis if a lineage was obtained from processing the swab taken for diagnosis, COVID-19 symptoms occurred on day 15 after the second dose or later, and before the participant was unblinded as to the vaccines they had received. In addition, some participants received a COVID-19 vaccine outside of the trial and were censored in the analysis at this time point.

Symptomatic cases occurring more than 21 days after a first dose but before the 15 day post-second dose timepoint were considered secondary endpoints for efficacy analyses.

Vaccine efficacy was defined as 100% x (1 – relative risk (RR)), where RR was estimated from an unadjusted robust Poisson model using SAS proc genmod. The log of the number of days of follow up was included as an offset in the model.

To determine if the SARS-CoV-2 lineage affected the viral load for the case, viral load data was compared across variants for swabs from cases included in the efficacy analysis, and separately from all processed swabs combined regardless of vaccines received. Viral loads were compared using the Kruskal-Wallis test.

Swabs were not available from all cases as some participants accessed NAAT tests at non-study sites and at one site a freezer malfunctioned. A sensitivity analysis was conducted using multiple imputations to impute the missing lineage data from unavailable swabs under a missing at random assumption. The imputation model generated a value from a three-component multinomial (categorical) variable in which the three components corresponded to 'Gamma (P.1) variant', 'Zeta (P.2) variant' or 'other variants'. The probabilities used to generate the imputed value were obtained from the site-specific distribution of Gamma (P.1) to Zeta (P.2) to other variants on the week the case occurred, by vaccine arm. This allowed the chronological and geographical spread of new variants to be incorporated into the imputation, and for any potential difference in efficacy by variant to be incorporated in the imputation model. 100 imputation datasets were generated and the estimate and its standard error stored for each iteration. The 100 imputed estimates were combined using Rubin's rules[41,42] in SAS proc mianalyze.

The data cut-off date for this analysis was February 28, 2021 at which point the majority of participants in the trial were unblinded and vaccinated. Follow up of continues, however cases that accrued after unblinding and vaccination do not contribute to efficacy analyses.

Data collection was done using RedCap version 9.5.22. Statistical analysis was done using SAS version 9.4 and R version 4.0.4. Bioinformatics analysis was conducted in Python with Pandas 0.25.3. Consensus sequences were aligned using MAFFT version 7.402.

The trial was conducted according to the principles of the Declaration of Helsinki and was approved by the Brazilian National research Ethics Committee (ref: 32604920.5.0000.5505), and the Oxford Tropical Research Ethics Committee (ref. [20–36]).

The trial is registered at ISRCTN89951424.

*RNA extraction and viral load quantification.* For 96% of the sample set (650/676 samples, 516/518 of efficacy cohort), RNA was extracted from primary samples shipped at −80 °C from participating sites in Brazil to the University of Oxford. The remaining samples were shipped as pre-extracted RNA. SARS-CoV-2 viral RNA was extracted from swab samples using the Quick-DNA/RNA Viral kit (Zymo Research): 200 μL of sample was mixed with 200 μL of DNA/RNA shield,

before being extracted according to the manufacturer's spin column protocol. RNA was eluted in 50 μL of DNAse/RNAse-free water and frozen at −80 °C. SARS-CoV-2 RNA was quantified by real-time polymerase chain reaction (RT-PCR) using the CDC N1 oligonucleotide set (https://www.cdc.gov/coronavirus/2019-ncov/lab/rt-pcr-panel-primer-probes.html) and the Quantitect Probe RT-PCR kit (QIAGEN) in a 25 μL reaction volume containing 2 μL of extracted RNA. Oligonucleotides (ATDBio) were resuspended in ultrapure water. RT-PCR was performed on an Applied Biosystems StepOnePlus Real-Time PCR system (ThermoFisher Scientific) with the following settings: 50 °C for 30 min (reverse transcription), 95 °C for 10 min (hot-start polymerase activation), and 40 cycles of 94 °C for 15 sec (denaturation) and 60 °C for 1 min (combined annealing and extension). Intra-assay variation was controlled through use of a standard curve of synthetic RNA control 19/304 (NIBSC https://www.nibsc.org/products/brm_product_catalogue/detail_page.aspx?CatId=19/304) serially diluted from 1,000 copies/reaction to 10 copies/reaction. RT-PCR Ct values were converted to copy number/reaction using the standard curve, and to international units/mL by the conversion rate provided by NIBSC for samples with known processing volumes.

**Sequencing.** Samples with Ct<31 were taken forward for veSEQ sequencing as previously described[43], using 30 μl RNA per sample as input volume and performing target capture on batches of 90 samples, alongside a series of quantification standards and positive and controls. Samples were demultiplexed using unique dual indexes (UDI), and read output was validated against Ct values to confirm sample integrity. Genomes were assembled from sequencing reads using the ShiverCovid pipeline v1.8 (https://github.com/BDI-pathogens/ShiverCovid) with variant frequencies calculated using shiver (tools/AnalysePileup.py)[44], using default settings of no base alignment quality and maximum pileup depth of 1000000. Lineages were assigned by Pangolin version 2.4.2 (lineages version 2021–04–28) combined with phylogenetic placement within the relevant clade, using the determined consensus genome for each sequenced sample. For incomplete genomes, lineages were assigned based on presence of lineage-defining mutations for Gamma (P.1) and Zeta (P.2) in the sequencing reads (https://github.com/phe-genomics/variant_definitions/blob/main/variant_yaml/) and by genotyping as described below.

**Phylogenetic reconstruction.** Consensus sequences were aligned using MAFFT version 7.402[45] with the default settings (algorithm FFT-NS-2, 6merpair, retree 2, weighting factor 2.7, gap opening 1.53, gap extension 0.123). Phylogenetic reconstruction was performed on the alignment consisting of consensus sequences rooted with the Wuhan-Hu-1 reference sequence (RefSeq NC_045512), using IQ-TREE version 1.6.12[46], with the generalised time reversible + FreeRate model and 1000 bootstrap replicates.

**Genotyping.** Samples for which genome sequencing did not give a clear lineage classification, or which showed evidence of RNA degradation (as identified by unexpectedly low read yield and library fragment sizes <200b; typical median fragment size 380b), were genotyped using allele specific PCR (ASP)-based assays[47]. Custom Gamma (P.1) and Zeta (P.2) ASP assays were designed to identify lineage-specific and highly sensitive single-nucleotide polymorphisms (SNPs) S:K417T (Gamma, P.1) and ORF1a:L3468V (Zeta, P.2). The ASP utilizes two dye-labelled probes that differ only in the SNP location, and leverages differential binding affinities of each probe due to primer-target mismatches to genotype the SNP with a higher sensitivity than sequencing. The assays were validated using sequence-confirmed Gamma (P.1) and Zeta (P.2) samples from the present dataset, with samples from other non-Gamma/Zeta (P.1/P.2) lineages as controls (Supp. Figs. 1 and 2). ASP was performed using the Quantitect Probe RT-PCR kit (QIAGEN) in a 25 μL reaction volume containing 5 μL of extracted RNA and performed on the Applied Biosystems StepOnePlus Real-Time PCR system using a genotyping program. Gamma (P.1) and Zeta (P.2) oligonucleotide sequences and reaction concentrations are listed in Supplementary Table 5. The P.1 ASP was performed with the following settings: 50 °C for 30 min, 60 °C for 30 seconds (pre-amplification read), 95 °C for 10 min, 45 cycles of 95 °C for 15 sec, 58 °C for 20 seconds, and 60 °C for 45 seconds, and 60 °C for 30 seconds (post-amplification read). The Zeta (P.2) ASP was performed with the following settings: 50 °C for 30 min, 66.5 °C for 30 seconds (pre-amplification read), 95 °C for 10 min, 50 cycles of 95 °C for 15 sec and 66.5 °C for 1 minute, and 66.5 °C for 30 seconds (post-amplification read). cDNA of sequence-confirmed samples was generated using the SuperScript III First-Strand Synthesis System (ThermoFisher Scientific) according to the manufacturer's instructions for gene-specific primers, except reverse transcription of the Zeta (P.2) cDNA controls was performed at 50 °C. Serially diluted cDNA aliquots of sequence-confirmed Gamma (P.1), Zeta (P.2), and non-Gamma/Zeta (P.1/P.2) samples were used as discrimination controls and ultrapure water served as no-template controls (NTCs). The change in fluorescent signal between pre-amplification and post-amplification reads for both dye-labelled probes was plotted on a cartesian plane. SNPs were designated based on their clustering with discrimination controls. Samples that failed to achieve a change in signal in either probe greater than those of the NTCs or lacked evidence of amplification were designated "undetermined." Samples that were genotyped as non-Gamma/Zeta (P.1/P.2) by ASP and had no sequence data were classified as "Other lineage (non-

Gamma/Zeta, P.1/P.2)". Samples that could not be assigned a lineage by either sequencing or genotyping were classified as "Undetermined".

**Reporting summary**. Further information on research design is available in the Nature Research Reporting Summary linked to this article.

## Data availability

Anonymised participant data will be made available when the trial is complete, upon requests directed to the corresponding author. Proposals will be reviewed and approved by the sponsor, investigator, and collaborators on the basis of scientific merit. After approval of a proposal, data can be shared through a secure online platform after signing a data access agreement. All data will be made available for a minimum of 5 years from the end of the trial.

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

## Acknowledgements

This Article reports independent research supported by the Wellcome Trust Core Award Grant Number 203141/Z/16/Z, UK Research and Innovation, National Institute for Health Research (NIHR), The Coalition for Epidemic Preparedness Innovations, the Lemann Foundation, Rede D'Or, the Brava and Telles Foundation, and AstraZeneca. We acknowledge support from the NIHR Oxford Health Biomedical Research Centre. PMF

received funding from the Coordenacao de Aperfeicoamento de Pessoal de Nivel Superior, Brazil (finance code 001). The views expressed in this publication are those of the authors and not necessarily those of the NIHR or the UK Department of Health and Social Care. We thank the volunteers who participated in this study. AstraZeneca reviewed the final manuscript before submission, but the academic authors retained editorial control. All other funders of the study had no role in study design, data collection, data analysis, data interpretation, or writing of the report. UKRI, NIHR, CEPI, Wellcome Trust, the Lemann Foundation, Rede D'OR, the Brava and Telles Foundation, NIHR Oxford Biomedical Research Centre, and AstraZeneca.

## Author contributions

A.J.P. and S.C.G. conceived the trials and A.J.P. is the chief investigator. A.J.P., S.A.C.C., L.Y.W., D.J., K.R.W.E., M.N.R., M.V., P.M.F., T.G. contributed to the protocol and design of the study. S.A.C.C., L.Y.W., A.V.D.A.M., E.P.M., A.P., A.V.S., E.S. are study site principal investigators. P.K.A., E.P., D.J., P.M.F., S.B., M.F., S.K., S.Ke, Y.F.M., contributed to the implementation of the study or data collection. N.G.M., and M.V. did the statistical analysis. T.G., D.B., J.R., P.S., C.F., conducted the sequencing and genotyping. A.J.P., T.G., K.R.W.E., M.N.R., T.L., and M.V. contributed to the preparation of the report. All authors critically reviewed and approved the final version.

## Competing interests

Oxford University has entered into a partnership with AstraZeneca for further development of ChAdOx1 nCoV-19. AstraZeneca reviewed the data from the study and the final manuscript before submission, but the authors retained editorial control. S.C.G. is cofounder of Vaccitech (collaborators in the early development of this vaccine candidate) and named as an inventor on a patent covering use of ChAdOx1-vectored vaccines (PCT/GB2012/000467) and a patent application covering this SARS-CoV-2 vaccine. P.M.F. is a consultant to Vaccitech. A.J.P. is Chair of the UK Department of Health and Social Care's JCVI, but does not participate in policy advice on coronavirus vaccines, and is a member of the WHO Strategic Advisory Group of Experts (SAGE). A.J.P. is a NIHR Senior Investigator. All other authors declare no competing interests.

## Additional information

## the AMPHEUS Project

David Buck[17], Angie Green[17], George MacIntyre-Cockett[17], Paolo Piazza[17], John A. Todd[17], Amy Trebes[17] & Laura Thomson[18]

[18]Oxford Viral Sequencing Group, Big Data Institute, University of Oxford, Oxford, UK.

## Oxford COVID Vaccine Trial Team

Lygia Accioly Tinoco[19], Karla Cristina Marques Afonso Ferreira[20], Cenusa Almeida[19], Brian Angus[3], Beatriz Arns[21], Laiana Arruda[19], Renato De Ávila Kfouri[22], Lucas Henrique Azevedo da Silva[20], Matheus José Barbosa Moreira[20], Brenda Vasconcelos Barbosa Paiva[9,10], Louise Bates[1], Nancy Bellei[22], Bruno Boettger[22], Leandro Bonecker Lora[9,10], Nina Amanda Borges de Araújo[20], Chrystiane do Nascimento Brito de Oliveira[20], Charlie Brown-O'Sullivan[3], Daniel Calich Luz[20], Joao Renato Cardoso Mourão[9,10], Caroline Scherer Carvalho[23], Paola Cicconi[3], Ana Gibertoni Cruz[24], Debora Cunha[21], Daniel Marinho Da Costa[9,10], Isabela Garrido Da Silva Gonzalez[22], Priscila de Arruda Trindade[25], Bruno Solano de Freitas Souza[26], Sergio Carlos Assis De Jesus Junior[9,10], Maria Isabel de Moraes Pinto[22], Karolyne Porto De Mores[9,10], Maristela Miyamoto de Nobrega[22], Milla Dias Sampaio[19], Janaína Keyla Dionísio dos Santos[20], Alexander D. Douglas[3], Suzete Nascimento Farias da Guarda[19,27], Mujtaba Ghulam Farooq[1], Shuo Feng[1], Marcel Catão Ferreira dos Santos[20], Marília Miranda Franco[28], Marianne Garcia de Oliveira[20], Fernanda Garcia Spina[22], Tannyth Gomes dos Santos[20], Alvaro Henrique Goyanna[9,10], Rosana Esteves Haddad[29], Adrian V. S. Hill[3], Mimi M. Hou[3], Bruna Junqueira[19], Bruna Somavilla Kelling[30], Baktash Khozoee[3], Renan Gustavo Kunst[23], Jonathan Kwok[31], Meera Madhavan[3],

José Antônio Mainardi de Carvalho[25], Olga Mazur[1], Angela M. Minassian[3], Leonardo Motta Ramos[9,10], Celia Hatsuko Myasaki[22], Helena Carolina Noal[32], Natália Nóbrega de Lima[20], Rabiullah Noristani[1], Ana Luiza Perez[21], Daniel J. Phillips[1], Priscila Pinheiro[19], Jéssica Morgana Gediel Pinheiro[21], Marie Marcelle Prestes Camara[20], Isabella Queiroz[19], Alessandra Ramos Souza[22], Thais Regina Y. Castro[30], Hannah Robinson[1], Marianna Rocha Jorge[19], Talita Rochetti[22], Mariana Bernadi S. Saba[22], Natalia Zerbinatti Salvador[9,10], Fernanda Caldeira Veloso Santos[23], Mayara Fraga Santos Guerra[9,10], Samiullah Seddiqi[1], Roberta Senger[23], Robert Shaw[1], Airanuedida Silva Soares[20], Rinn Song[1], Guilherme G. Sorio[21], Ricardo Stein[21], Arabella V. S. Stuart[1], Tais Tasqueto Tassinari[32], Cheryl Turner[3], Tarsila Vieceli[21], Taiane A. Vieira[21], João Gabriel Villar Cavalcanti[20], Marion E. E. Watson[3], Andy Yao[1] & Rafael Zimmer[21]

[19]Hospital São Rafael e ID'OR, Porto Alegre, Brazil. [20]Centro de Estudos e Pesquisas em Moléstias Infecciosas, Rio de Janeiro, Brazil. [21]Hospital de Clinicas de Porto Alegre, Porto Alegre, Brazil. [22]Universidade Federal de Sao Paulo, São Paulo, Brazil. [23]Hospital Universitário de Santa Maria, Santa Maria, Brazil. [24]Nuffield Department of Population Health, University of Oxford, Oxford, UK. [25]Department of Clinical and Toxicological Analysis-Universidade Federal de Santa Maria, Santa Maria, Brazil. [26]Monte Tabor Centro Ítalo Brasileiro de Promoção Sanitária, Fiocruz-BA e I'DOR, Rio de Janeiro, Brazil. [27]Universidade Federal da Bahia, Salvador, Brazil. [28]Rede D'OR, Rio De Janeiro, Brazil. [29]Vaxtrials, F&F Tower, Calle 50, Panamá, Panama. [30]Universidade Federal de Santa Maria, Santa Maria, Brazil. [31]Nuffield Department of Medicine, University of Oxford, Oxford, UK. [32]Programa de Pós Graduação em Enfermagem- PPGENF - Universidade Federal de Santa Maria, Santa Maria, Brazil.

