## [Peer Review File · Nature Communications]

Efficacy of ChAdOx1 nCoV-19 (AZD1222) vaccine against SARS-CoV-2 lineages circulating in BrazilReviewers' Comments:

Reviewer #1:

Remarks to the Author:

Despite being an important study, due to the number of outcomes per variant, it is difficult to compare the effectiveness for all variants. It is important to rewrite the manuscript with the correct VOC names (Alpha, Beta, Gamma, Zeta, etc.)

The main finding concerns the efficacy of ChAdOx1 nCoV-19 against the P2 variant which appears to be similar to the B.1.1.28 variant.

Importantly, the authors determine the efficacy for different variants according to the number of doses (D1 and D2). Currently, Brazil uses a 12-week interval between doses and understanding the efficacy and/or effectiveness of ChAdOx1 nCoV-19 for one dose and two doses in the context of the Gamma variant is very relevant. Several manuscripts and pre-prints have demonstrated the need for 2 doses of ChAdOx1 nCoV-19 to preserve vaccine effectiveness for the new variants (especially the Gamma and Delta variants). Therefore, I suggest adding this comparative analysis as was done in the clinical trial to see if there are important differences regarding the effectiveness of the first dose of ChAdOx1 for the different variants.

In the discussion it is not out of context to discuss CoronaVac. There are no published data, there is no data regarding the variants in the studies cited, and there is no direct relationship with the assessment of ChAdOx1 for new variants.

It would be important that efficacy data even after blind removal were calculated and discussed in the manuscript. There is only one pre-print that evaluated the effectiveness of ChAdOx1 nCoV-19 for the Gamma/Beta variant in Canada and for just one dose of ChAdOx1 nCoV-19. The efficacy and/or effectiveness information of ChAdOx1 nCoV-19 for the P1 variant is extraordinarily relevant despite the limitations. The authors should present not only the evaluation of 18 cases of Gamma variants identified before unblinding, but all 160 that occurred after unblinding. As all the analysis present is a post-hoc analysis, with some limitation regard to the power, I don't understand the reason for not include in the analysis this participants with the properly discuss the limitation of non-blinding in the estimation of efficacy.

Reviewer #2:

Remarks to the Author:

This is a well-written paper (except for an odd incomplete sentence on line 312). It addresses an important scientific question, namely how the efficacy differs for the different variants. I have a few detailed comments:

1. As shown in Table S2, only about one tenth of the P.2 cases are included in the efficacy analysis due to unblinding. For a secondary analysis of this efficacy trial, it seems that additional analyses of cases after unblinding with sensitivity studies are essential to answering the scientific question of interest.
2. Since the primary paper for the immunogenicity data has also been published, studying variants-specific immunogenicity and relate that to the efficacy seems valuable.
3. Will variants-specific efficacy results for other geographical regions be reported elsewhere? It would be worth briefly explaining what to expect in the paper.
4. In statistical analyses, what covariates are adjusted in the Poisson models?
5. How do the Poisson models address competing risk?

Reviewer #3:

Remarks to the Author:

Overall, this is important work and the paper has been carefully prepared. The statistical analysis plan is robust and uses appropriate statistical methodology. There are a few minor clarifications that are to be considered.

The inclusion of subjects and events that occurred after the approval of the vaccine is somewhat complex. Consider developing a figure that shows how the final inclusion / exclusion into the analysis set was implemented. Some text such as page 10, lines 202 – 204 would suggest that infections after the public availability / unblinding were not gathered. In a way, this is understandable from a pure study perspective, but from a public health perspective with emerging variants, one would hope all available data would be summarized. Said differently, if there were events that happened after unblinding, where they included in the active vaccine group? What about the crossed over patients? Were there any placebo vaccinated that were never crossed over—could this data be shown?

The paper makes what could be a strong statistical claim of “similar” response across the variants. The study was not designed to assess equivalence across variants and typical methods for equivalence designs were not implemented (e.g., a priori limits of equivalence). Most likely, the best path forward is to not interpret the data in the context of similarity and instead, just report the effectiveness of the vaccine in each group. If “similar” is desired as a construct, estimate the difference in effectiveness and its 95% CI. This can help the reader determine just how similar the effectiveness ratios are. It is expected that the confidence interval will be quite wide suggesting limited data on the equivalence across variants.

While the variants may capture the temporal changes in vaccine effectiveness, it may also be important to show some of the data using a few different time scales – time from vaccination; and infections by calendar day. This could be purely exploratory and posted as a supplement.

REVIEWER COMMENTS

Reviewer #1 (Remarks to the Author):

Despite being an important study, due to the number of outcomes per variant, it is difficult to compare the effectiveness for all variants. It is important to rewrite the manuscript with the correct VOC names (Alpha, Beta, Gamma, Zeta, etc.)

RESPONSE: WHO Greek names have been added as suggested and lineage retained in brackets as non-VOC/VUIs lineages are also included in manuscript.

The main finding concerns the efficacy of ChAdOx1 nCoV-19 against the P2 variant which appears to be similar to the B.1.1.28 variant.

Importantly, the authors determine the efficacy for different variants according to the number of doses (D1 and D2). Currently, Brazil uses a 12-week interval between doses and understanding the efficacy and/or effectiveness of ChAdOx1 nCoV-19 for one dose and two doses in the context of the Gamma variant is very relevant. Several manuscripts and pre-prints have demonstrated the need for 2 doses of ChAdOx1 nCoV-19 to preserve vaccine effectiveness for the new variants (especially the Gamma and Delta variants). Therefore, I suggest adding this comparative analysis as was done in the clinical trial to see if there are important differences regarding the effectiveness of the first dose of ChAdOx1 for the different variants.

RESPONSE: This has now been added as suggested as Supplementary Table 4. There are fewer cases occurring after a single dose than after two doses as there was only a short period of follow up before the second dose was received, and the cases occurred earlier in the trial. There is only 1 case of P.1. The P.2 variant is the only one with more than 10 cases. Efficacy analyses have not been performed due to the small numbers but the data are presented in tabular form by vaccine arm.

In the discussion it is not out of context to discuss CoronaVac. There are no published data, there is no data regarding the variants in the studies cited, and there is no direct relationship with the assessment of ChAdOx1 for new variants.

RESPONSE: We have interpreted this comment to mean the reviewer felt it was out of context to discuss CoronaVac and so this section has been removed from the discussion.

It would be important that efficacy data even after blind removal were calculated and discussed in the manuscript. There is only one pre-print that evaluated the effectiveness of ChAdOx1 nCoV-19 for the Gamma/Beta variant in Canada and for just one dose of ChAdOx1 nCoV-19. The efficacy and/or effectiveness information of ChAdOx1 nCoV-19 for the P1 variant is extraordinarily relevant despite the limitations. The authors should present not only the evaluation of 18 cases of Gamma variants identified before unblinding, but all 160 that occurred after unblinding. As all the analysis present is a post-hoc analysis, with some limitation regard to the power, I don't understand the reason for not include in the analysis this participants with the properly discuss the limitation of non-blinding in the estimation of efficacy.

RESPONSE: We agree on the importance and relevance of understanding the efficacy against P.1. When participants in the trial were unblinded, the control group were vaccinated with ChAdOx1 nCoV-19. Therefore there is now no longer a control group that can be used to calculate efficacy. Instead we have two groups that have both been vaccinated with the same vaccine. It is very unfortunate that this data is not useful to inform us on the efficacy of the vaccine against P.1 due to the timing of this variant arriving in our study sites. We have added the Canadian preprint to the discussion and thank the reviewer for this suggestion.

Reviewer #2 (Remarks to the Author):

This is a well-written paper (except for an odd incomplete sentence on line 312). It addresses an important scientific question, namely how the efficacy differs for the different variants. I have a few detailed comments:

RESPONSE: Thank you for pointing out the typo in line 312 which has now been amended.

1. As shown in Table S2, only about one tenth of the P.2 cases are included in the efficacy analysis due to unblinding. For a secondary analysis of this efficacy trial, it seems that additional analyses of cases after unblinding with sensitivity studies are essential to answering the scientific question of interest.

RESPONSE: We agree that this is an important scientific question and would like to have included all the cases. When participants in the trial were unblinded, the control group were vaccinated with ChAdOx1 nCoV-19. Therefore there is now no longer a control group that can be used to calculate efficacy. Instead we have two groups that have both been vaccinated with the same vaccine. It is very unfortunate that this data cannot be used to inform us on the efficacy of the vaccine after unblinding.

2. Since the primary paper for the immunogenicity data has also been published, studying variants-specific immunogenicity and relate that to the efficacy seems valuable.

RESPONSE: We agree that this would be a valuable analysis. We have published some data on immune responses against P1 (<https://doi.org/10.1016/j.cell.2021.06.020>), but data are not currently available to relate variant-specific immunity to efficacy in a formal correlates analysis. We have added a comment about immunogenicity in the discussion

3. Will variants-specific efficacy results for other geographical regions be reported elsewhere? It would be worth briefly explaining what to expect in the paper.

RESPONSE: Yes these are published already. The efficacy results for South Africa (Beta) and UK (Alpha) are published in the NEJM and Lancet journals as follows:

<https://www.nejm.org/doi/full/10.1056/nejmoa2102214>

[https://www.thelancet.com/journals/lancet/article/PIIS0140-6736\(21\)00628-0/fulltext](https://www.thelancet.com/journals/lancet/article/PIIS0140-6736(21)00628-0/fulltext)

Both these papers are referred to in the paper. Arrival of Delta variant occurred after study unblinding and so further efficacy analysis for this variant is not possible, but real-world effectiveness has been reported.

4. In statistical analyses, what covariates are adjusted in the Poisson models?

RESPONSE: The Poisson model was unadjusted. This has been added to the methods section for clarity.

5. How do the Poisson models address competing risk?

RESPONSE: The Poisson model is not a competing risk model. Whilst it is possible that a second infection or other event may occur which would introduce a competing risk bias, there were only an extremely small numbers of participants with a second infection, and very good retention an follow up, therefore competing risk models were not implemented.

Reviewer #3 (Remarks to the Author):

Overall, this is important work and the paper has been carefully prepared. The statistical analysis plan is robust and uses appropriate statistical methodology. There are a few minor clarifications that are to be considered.

The inclusion of subjects and events that occurred after the approval of the vaccine is somewhat complex. Consider developing a figure that shows how the final inclusion / exclusion into the analysis set was implemented. Some text such as page 10, lines 202 – 204 would suggest that infections after the public availability / unblinding were not gathered. In a way, this is understandable from a pure study perspective, but from a public health perspective with emerging variants, one would hope all available data would be summarized. Said differently, if there were events that happened after unblinding, were they included in the active vaccine group? What about the crossed over patients? Were there any placebo vaccinated that were never crossed over—could this data be shown?

RESPONSE: Thank you for the suggestion. We have amended the consort diagram in Figure 1 to show the cases that occurred after the unblinding/revaccination of the control group, and the number of control group participants who did not accept vaccination at the time of unblinding or shortly thereafter (<10%). We have also updated the text in the methods page 10 to be more clear about the continuing follow up of cases after unblinding.

The paper makes what could be a strong statistical claim of “similar” response across the variants. The study was not designed to assess equivalence across variants and typical methods for equivalence designs were not implemented (e.g., a priori limits of equivalence). Most likely, the best path forward is to not interpret the data in the context of similarity and instead, just report the effectiveness of the vaccine in each group. If “similar” is desired as a construct, estimate the difference in effectiveness and its 95% CI. This can help the reader determine just how similar the effectiveness ratios are. It is expected that the confidence interval will be quite wide suggesting limited data on the equivalence across variants.

RESPONSE: We agree with the reviewer and have removed these references to similar efficacy across variants.

While the variants may capture the temporal changes in vaccine effectiveness, it may also be important to show some of the data using a few different time scales – time from vaccination;

and infections by calendar day. This could be purely exploratory and posted as a supplement.

RESPONSE: We have shown infections by calendar week in Figure 2A.

Reviewers' Comments:

Reviewer #1:

Remarks to the Author:

The authors answered all que questions and modified the text according to the suggestions. The manuscript has improved a lot mainly by making clear the limitations of the results in relation to power due to the reduced number of samples. Thus, I approve the current version for publications without additional comments.

Reviewer #2:

None

Reviewer #3:

Remarks to the Author:

The authors have amended the paper consistent with the prior review. No further concerns.